# Characteristics of astigmatism before and 1 month after blepharoptosis surgery in patients with acquired ptosis

**Kazuhiko Dannoue**[1]*, **Seiji Takagi**[1,2], **Keiko Uemura**[1], **Anna Takei**[1], **Tomohiko Usui**[1,3]

**1** Dannoue Eye Clinic, Kawasaki, Japan, **2** Department of Ophthalmology, Toho University Oomori Medical Hospital, Ota City, Tokyo, **3** Department of Ophthalmology, Graduate School of Medicine, International University of Health and Welfare, Tokyo, Japan

* kz_dannoue@yahoo.co.jp

## Abstract

In this study, we aimed to evaluate the characteristics of astigmatism preoperatively and 1 month postoperatively in patients with age-related ptosis (AP) and contact lens-related ptosis (CLP), and investigate surgery-induced astigmatism (SIA) using the Jaffe vector analysis and the Cravy method. Consecutive patients who underwent blepharoptosis surgery between January 2019 and December 2019 were included. The patients were divided into AP and CLP groups. Computerized corneal topography was used to assess the magnitude and axis of corneal astigmatism. Astigmatism was classified as with-the-rule (WTR), against-the-rule (ATR), or oblique astigmatism (OA) pre- and postoperatively. SIA was calculated by vector analysis using the Cravy and Jaffe methods. The correlation between SIA and margin reflex distance (MRD) was calculated. One hundred and eight eyes from 58 patients (AP group: 85 eyes from 45 patients, CLP group: 23 eyes from 13 patients) were included. The AP group (73.8±7.6 years) was significantly older than the CLP group (47.7 ±6.6 years). The MRD increased significantly after treatment in both groups. The proportions of WTR, ATR, and OA were 52%, 22%, and 25%, and 86%, 9%, and 4% in the AP and CLP groups, respectively. A shift in astigmatism type was observed in 41% and 13% of patients in the AP and CLP groups, respectively. The average SIA measured using the Cravy method was 0.11±1.22 D in the AP group and −0.28±1.07 D in the CLP group (WTR astigmatism). The SIA calculated using the Jaffe method was 0.78±0.70 D in the AP group and 0.82±0.88 D in the CLP group. There was no significant correlation between SIA calculated using the Cravy and Jaffe methods and MRD. ATR was most common in age-related ptosis and WTR was most common in contact lens-related ptosis. Upper eyelid re-positioning may affect visual functions due to astigmatic changes in the short term postoperatively.

## Introduction

Acquired ptosis is a common ophthalmological condition and is associated with aging [1], prolonged use of hard contact lens [2], and anterior segment surgery [3]. Acquired ptosis does not

**Data Availability Statement:** All relevant data are within the paper and its Supporting information files.

**Funding:** The authors received no specific funding for this work.

**Competing interests:** NO The authors have declared that no competing interests exist.

just cause esthetic problems but also impairment of visual functions. Altitudinal visual field defects that impair the upward gaze and are measured using manual kinetic perimetry by the Goldmann perimeter [4] or static automated perimetry [5] have been well documented. Limiting of light influx due to palpebral narrowing could cause impairment of contrast sensitivity [6]. Moreover, a retracted eyelid affects the shape of the cornea as the upper eyelids are directly adhered to the cornea [7]. Previous studies have reported on the influence of eyelids on the corneal surface in various eyelid conditions, including chalazion [8], hemangiomas [9], downward gaze [10], and ptosis [11], using localized topographical analysis.

The corneal surface plays an important role in refraction. Hence, increased pressure on the upper eyelid could result in corneal refractive changes, which could cause amblyopia [12] in congenital ptosis and persistent blurred vision in acquired ptosis [13]. Blepharoplasty is performed to correct the upper eyelid position. Various studies have reported astigmatic recovery after upper eyelid surgery [13–19]; however, patients who underwent blepharoptosis surgery occasionally complained of blurred vision in the operated eye [20]. The other astigmatic change observed after blepharoptosis surgery is change in the axis of the cylinder. Classically, the mechanical effect of ptosis may result in a steep corneal curvature in the vertical meridian, i.e., with-the-rule (WTR) corneal astigmatism [13,16]. However, a limited number of studies have reported on the astigmatic axis pre- and postoperatively for acquired ptosis [16]. Surgery-induced astigmatism should be assessed in both direction and magnitude, calculation methods for which have been discussed and reported, especially following cataract surgery [21–28]. In this study, we aimed to evaluate the characteristics of astigmatism in patients with two representative acquired ptosis (age-related and contact lens-related ptosis), preoperatively and 1 month postoperatively, and investigate blepharoptosis surgery-induced astigmatism (SIA) using the Jaffe vector analysis [21] and the Cravy method [22]. Moreover, the correlation among SIA assessed using the Jaffe method, that using the Cravy method, and the patient background was analyzed.

## Patients and methods

### Ethics

The Dannoeu Eye Clinic Ethics Committee reviewed and approved the protocol of this retrospective observational study and waived the requirement for informed consent with an opt-out display in the hospital. This study conformed to the tenets of the Declaration of Helsinki.

### Patients

Consecutive patients with acquired ptosis (AP) (age, >60 years; no history of any intraocular surgery and prolonged contact lens wear) and contact lens-related ptosis (CLP) (age, <59 years; long history of hard contact lens wear) who underwent blepharoptosis surgery between January 2019 and December 2019 were included. On the last visit before surgery, all patients underwent complete ocular examinations, including best-corrected visual acuity (BCVA), slit-lamp biomicroscopy, dilated fundoscopy, and optical coherence tomography. BCVA was obtained using Landolt C charts. These values were then converted to the logarithm of the minimum angle of resolution equivalent for statistical comparisons. Other potentially confounding retinal pathologies were excluded, and patients with a levator function of <4 mm were excluded. Ptosis severity was evaluated using the margin reflex distance (MRD), a measurement of the distance from the middle upper eyelid to the corneal light reflex [29]. Levator function was evaluated as eyelid excursion; upper lid was measured from the maximum downward gaze to the maximum upward gaze, with the frontalis muscle blocked by the examiner.

## Blepharoptosis surgery

Blepharoptosis surgery was conducted by one surgeon (KD) mainly using a $CO_2$ laser using the Müller's muscle tracking method. All surgeries were performed under local anesthesia using 2% xylocaine. In blepharoplasty, the excess skin is excised after tracing the skin that needs to be removed. After the tarsal plate was exposed, the levator brevis aponeurosis and the Müller's muscle were identified and detached carefully. Two or three places were tacked and crimped using 6–0 nylon, and the skin was sutured with a 7–0 nylon thread.

## Astigmatism

Corneal topography was measured using a computed topography system with a Scheimpflug camera (TMS5 Tomey, Japan) to measure the corneal elevation and the surface curvature. The patients were asked to blink twice to obtain reproducible measurements. To avoid any pressure from the eyelids on the corneal surface, the examination was performed with the patients' eyes open. Measurements were taken pre- and 1 month postoperatively when wound healing was observed. The refractive power was calculated from the radius of curvature, where the center of curvature was defined on the optical axis. The color map was constructed from one of the four scans with the least data variation.

Astigmatism was classified as follows: WTR astigmatism: steepest meridian close to the vertical meridian (30˚ on either side of the 90˚ meridian); "against the rule" (ATR) astigmatism: steepest meridian close to the horizontal meridian (30˚ on either side of the 180˚ meridian); and "oblique astigmatism" (OA): steepest meridians not close to either side of the vertical or horizontal meridians, within the aforementioned range but perpendicular to each other.

The degree of blepharoptosis SIA was calculated using Jaffe's vector analysis [21] and Cravy's vector analysis [22]. The following equations were used for each vector analysis:

*Jaffe method*

$$SIA(D) = \sqrt{|C_1^2 + C_2^2 - 2C_1C_2\cos2(A_1 - A_2)|}$$

$$SIA(Ax) = 0.5\tan^{-1}\frac{C_2\sin2A_2 - C_1\sin2A_1}{C_2\cos2A_2 - C_1\cos2A_1}$$

C1: preoperative cylinder, C2: postoperative cylinder, A1: axis of the cylinder of C1, A2: axis of the cylinder of C2.

*Cravy method*:

$$\text{Cravy}\Delta K = (\Delta x) \, 1 \, (\Delta y)$$

$$\Delta x = |K2\cos A2 - K1\cos A1|$$

$$\Delta y = |K2\sin A2 - K1\sin A1|$$

K1: preoperative cylinder, K2: postoperative cylinder.
A1: strong meridian of K1, A2: strong meridian of K2.

## Statistical analyses

All statistical analyses were performed using SPSS®, version 21 (SPSS Science, Chicago, IL, USA). The results of the descriptive analyses are reported as mean ± standard deviation. Astigmatism was compared using the two-tailed Mann–Whitney U test. The preoperative and

postoperative topographic parameters and BCVA were compared using Student's t-tests. The association among the individual SIA was calculated using the Cravy method, the Jaffe method, and the preoperative MRD. Statistical significance was set at P <0.05.

## Results

The demographic data of the patients are presented in Table 1. One hundred and eight eyes from 58 patients (AP group: 85 eyes from 45 patients; females, 32; age, 73.8±7.6 years; CLP group: 23 eyes from 13 patients; females, 10; age, 47.7±6.6 years) were included. There was a significant difference in age, though not in sex distribution, between the two groups. All participants were Asians. The MRD increased significantly after treatment in both the AP group (0.78±0.9 mm to 3.22±1.2 mm; P<0.001) and CLP group (0.70±0.5 mm to 3.85±0.5 mm; P<0.001). No significant change was observed in the visual acuity postoperatively in both groups.

The proportions of WTR, ATR, and OA were 52%, 22%, and 25% and 86%, 9%, and 4% in the AP and CLP groups, respectively (Fig 1).

The shift in the type of astigmatism postoperatively is shown in Table 2. In the AP group, 41% of patients showed a shift in the type of astigmatism. WTR changed to OA in 23.8% and ATR in 4.8% of the patients, OA changed to WTR in 26.4% and ATR in 26.3% of the patients, and ATR changed to WTR in 11.8% and OA in 31.8% of the patients. In the CLP group, 13% of patients showed a shift in the type of astigmatism. WTR changed to ATR in 4.8%, OA changed to WTR in 100%, and ATR changed to OA in 100% of the patients.

The distribution of blepharoptosis SIA in the AP and CLP groups measured using the Cravy method is shown in Fig 2. The average SIA measured using the Cravy method was a mean WTR astigmatism of 0.11±1.22 D in the AP group and −0.28±1.07 D in the CLP group. In the AP group, 58% of patients showed a WTR shift and 42% showed an ATR shift. In the CLP group, 52% patients showed a WTR shift and 48% showed an ATR shift. We observed that the astigmatic axis changed significantly (>1.00 D) in some patients in both groups.

The SIA measured using the Jaffe method was 0.78±0.70 D in the AP group and 0.82±0.88 D in the CLP group (P>0.05) (Fig 3). In 25% of patients, the SIA was >1.00 D as measured using the Jaffe method. No significant difference was observed between patients with mild and severe ptosis in the AP group (0.77±0.71 D vs. 0.8±0.75 D, P>0.05).

The correlation between SIA measured using the Cravy and Jaffe methods method and MRD did not show a significant relationship in the AP and CLP groups.

The representative cases are shown in Fig 4.

**Table 1. Demographic data of the age-related ptosis (AP) and contact lens-related ptosis (CLP) groups.**

| | AP | | | CLP | | | P-value |
|---|---|---|---|---|---|---|---|
| Number of eyes | 85 | | | 23 | | | - |
| Age (years) | 73.8±7.6 | | | 47.7±6.6 | | | <0.01* |
| Sex (male:female) | 13:32 | | | 3:10 | | | >0.05** |
| | Pre-operative value | Post-operative value | P-value*** | Pre-operative value | Post-operative value | P-value*** | - |
| MRD (mm) | 0.78±0.92 | 3.22±1.15 | <0.001 | 0.70±0.51 | 3.85±0.53 | <0.001 | - |
| Visual acuity (logMar) | 0.05±0.92 | 0.04±0.89 | >0.05 | −0.08±1.00 | −0.08+1.15 | >0.05 | - |

*Mann–Whitney U test,

**chi-square test, and

***student's t-test.

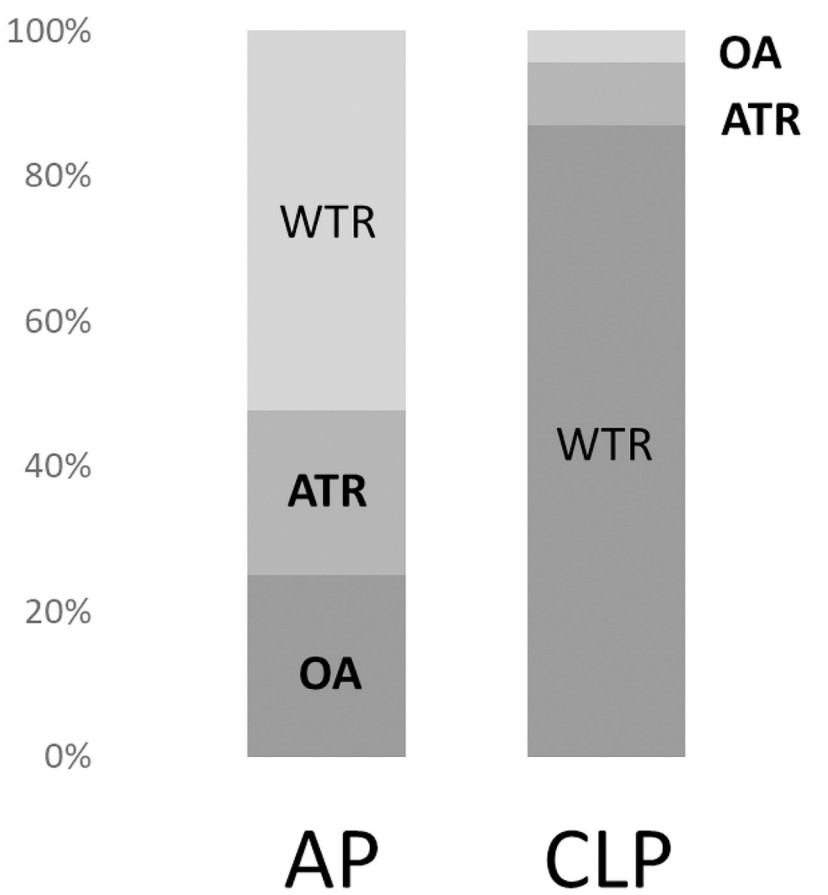

**Fig 1. Proportions of the preoperative astigmatism type in the acquired ptosis (AP) and contact lens-related ptosis (CLP) groups.** WTR: With-the-rule; ATR: Against the rule; OA: Oblique astigmatism.

In case 1, steep asymmetric deformation of the cornea was observed preoperatively in the upper part of the cornea, with improvement in corneal astigmatism after the surgical correction of ptosis; however, the astigmatic axis showed a change of about 30˚.

In case 2, oblique astigmatism was observed preoperatively and postoperatively, with increased degree of astigmatism 1 month postoperatively. In this case, the preoperative

**Table 2. Shift in the type of astigmatism postoperatively in the acquired ptosis (AP) and contact lens-related ptosis (CLP) groups.**

| AP | | CLP | |
|---|---|---|---|
| Preoperative (cases) | Postoperative (cases/%) | Preoperative (cases) | Postoperative (cases/%) |
| WTR (21) | WTR (15/71.4) | WTR (20) | WTR (1/95.0) |
| | OA (5/23.8) | | OA (0/0) |
| | ATR (1/4.8) | | ATR (1/5.0) |
| OA (19) | OA (9/47.3) | OA (2) | OA (0/0) |
| | WTR (5/26.3) | | WTR (2/100) |
| | ATR (5/26.3) | | ATR (0) |
| ATR (44) | ATR (25/56.8) | ATR (1) | ATR (0) |
| | OA (14/31.8) | | OA (100) |
| | WTR (5/11.3) | | WTR (0) |

The shaded cells represent a change in the type of astigmatism postoperatively.

WTR: With-the-rule; ATR: Against the rule; OA: Oblique astigmatism.

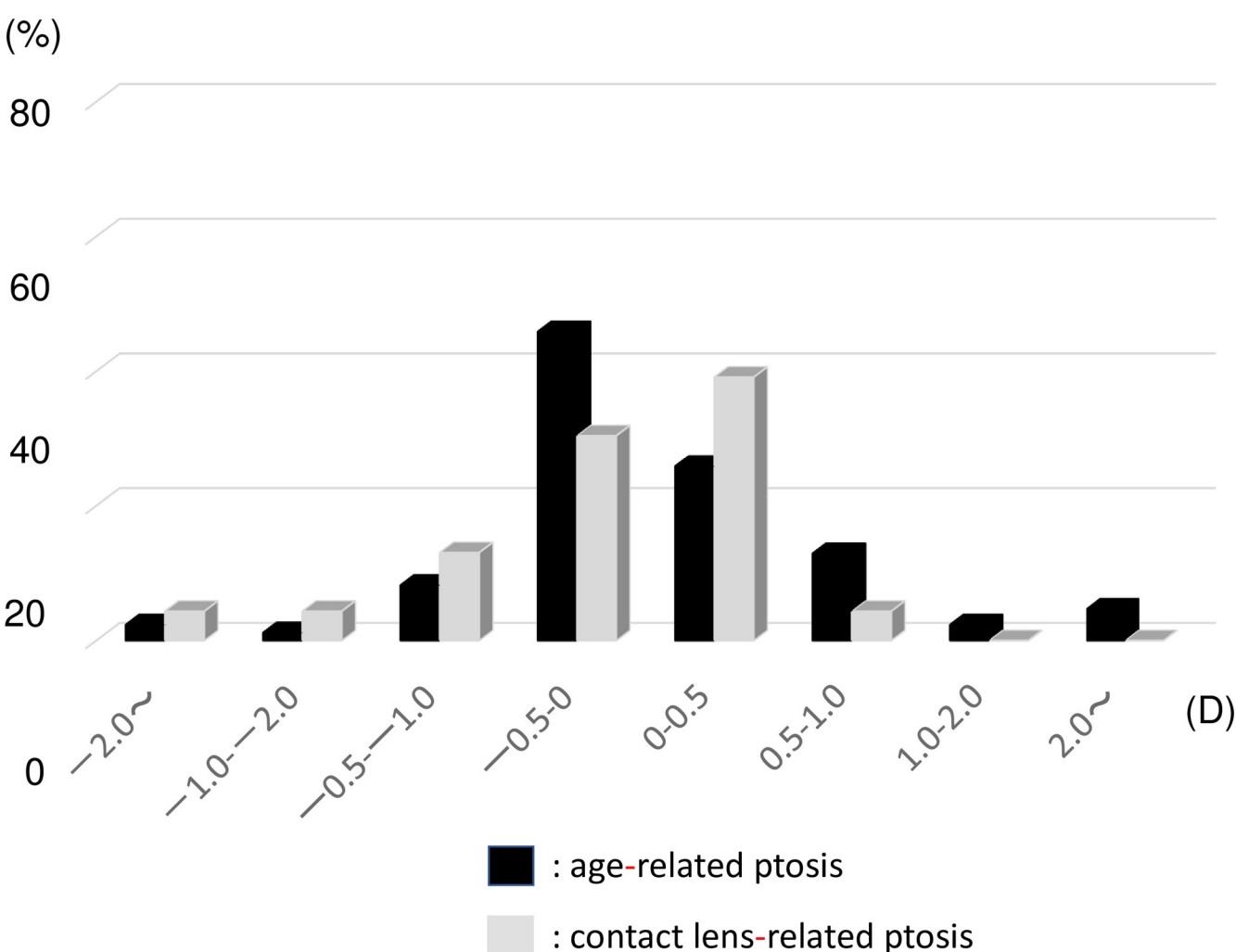

**Fig 2. Distribution of blepharoptosis surgery-induced astigmatism as measured using the Cravy method in the age-related ptosis group (black bar) and the contact lens-related ptosis groups (gray bar).** D: Diopter.

visual acuity was VS = 0.9 (1.2 × +0.75cly-0.75A120), and the postoperative visual acuity was VS = 0.5 (1.2 × +1.00 × -1.75A130), indicating a decrease in uncorrected visual acuity.

## Discussion

A retracted eyelid could affect the corneal morphology pre- and postoperatively, and this deformation could affect visual functions as a result of the astigmatic change. We focused on the change in astigmatic axis and calculated SIA using the Cravy and Jaffe methods. We observed that the type of astigmatism was different between the AP and CLP groups. The proportion of ATR was increased in both groups; in the AP group, the proportions of WTR and OA had increased, while in the CLP group, the proportion of WTR was high. Previously, it has been assumed that the mechanical pressure from ptosis of the upper eyelid causes the steepest corneal curvature in the vertical meridian, which leads to WTR drift. Gullstrand has reported that flattening of the peripheral cornea by eyelid pressure causes corneal astigmatism in the WTR direction [30]. We have also come across several cases in which the topography showed a strong asymmetric corneal astigmatism on the superior corneal hemifield preoperatively.

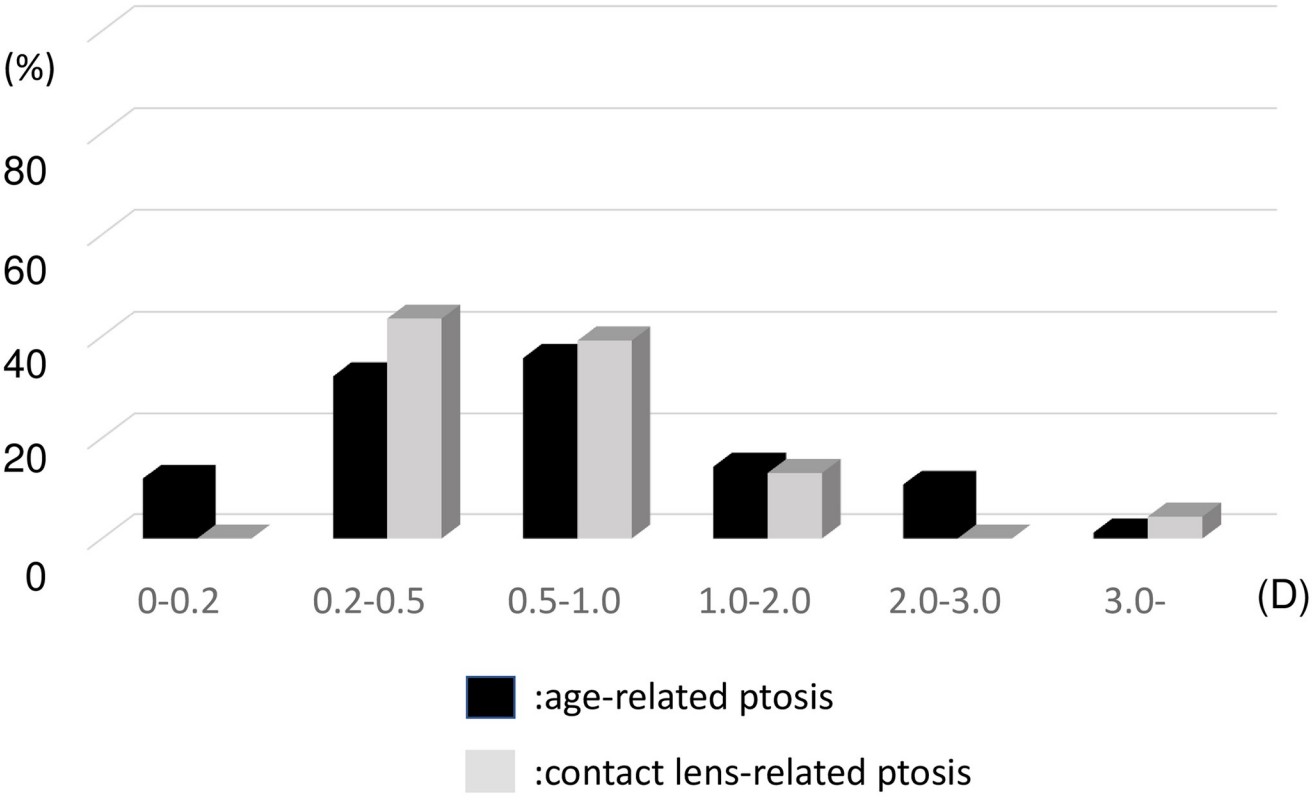

**Fig 3. Distribution of blepharoptosis surgery-induced astigmatism measured using the Jaffe method in the age-related ptosis and contact lens-related ptosis groups.** D: Diopter.

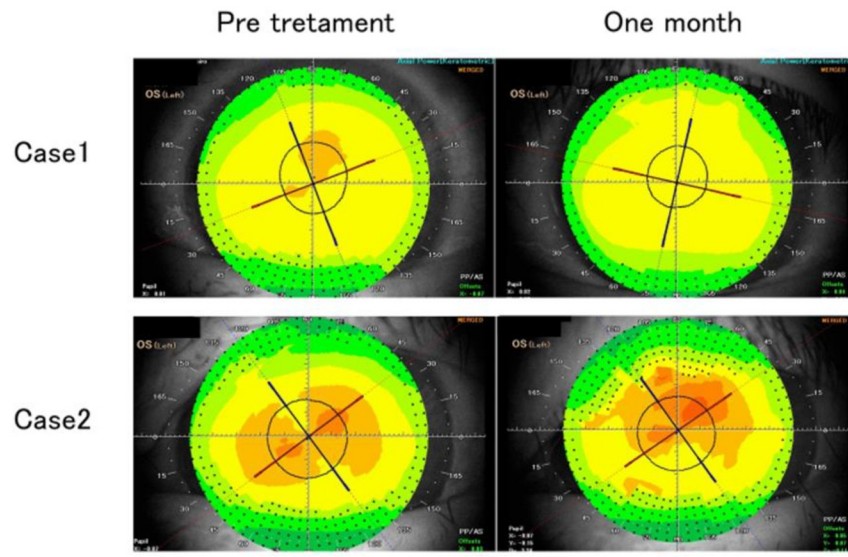

**Fig 4.** Case 1: Postoperative topography showing change in axis with improvement in focal steepening in the superior corneal hemisphere. Case 2: Postoperative topography showing increased magnitude of astigmatism and decreased uncorrected visual acuity 1 month postoperatively.

WTR was most frequent in the CLP group, while the proportions of WTR and OA were 25% and 53%, respectively, in the AP group. A possible reason for this difference might have been related to the age difference between the two groups. A previous study that investigated corneal astigmatism using videokeratometry in two different age groups reported that the preoperative proportion of WTR was 70% in the younger ptosis surgery group and 22% in the older blepharoptosis group [16], which is consistent with our results. Age-related changes in astigmatism in the normal population have been reported. Namba et al. reported that age-related variations in the corneal geometry showed horizontal steepening and vertical flattening of the corneal surface with increasing age in a large Japanese cohort study [31]. This age-related mechanism could significantly affect a large WTR proportion in the young CLP group and ATR proportion in the older AP group pre- and postoperatively.

Changes in the degree of astigmatism after blepharoptosis surgery have been reported previously [14–19]; however, changes in the astigmatic axis after blepharoptosis surgery have been reported in only a few studies [13,16]. This is the first study to report on the astigmatic shift after blepharoptosis using the Cravy and Jaffe method.

The method of calculating the astigmatic change, especially following cataract surgery, has been reported by Jaffe and Clayman [21], Cravy [22], Holladay [23], Olsen [24], Naeser [25,26] and others [27,28]. The Jaffe analysis uses the polar value method that is conceptually based on the surgically induced flattening and torque of the preoperative cylinder. In contrast, the Cravy method is a more mathematical method that focuses on the axial variation rather than the cylindrical; a positive value represents a WTR shift, while a negative value indicates an ATR shift [22]. The AP group showed a slight WTR shift, and the CLP group showed a slight ATR shift. This result might have been normal since the AP group had more patients with ATR and the CLP group had more patients with WTR preoperatively. The average change in diopter was −0.28 D and +0.11 D in the AP and CLP groups, respectively, which was minimal and not clinically significant. However, we also observed nine patients (11%) in the AP group and two patients (8%) in the CLP group who showed a significant astigmatic change of >1.00 D. We observed similar results in terms of the type of astigmatism: 40% of patients in the AP group and 17% of patients in the CLP group showed a change in the type of astigmatism postoperatively (Table 2). These results indicate that a large change in the astigmatic axis develops at a constant rate at 1 month postoperatively.

Brown et al. reported that persistent astigmatic changes were observed in ~10% of patients with ptosis and blepharoptosis [16]. This result is consistent with that of ours. The significant change in the astigmatic axis might have been due to the release of downward pressure on the cornea from the retracted contacted eyelid. However, we could not find a correlation between MRD and SIA measured using the Cravy and Jaffe methods. This result indicates that the relationship between astigmatism and the ptosis severity in the short term may not be linear but multifactorial. We speculated that one of the factors affecting this may be postoperative eyelid swelling. The eyelids are histologically prone to swelling and, occasionally, significant swelling after ptosis surgery may put pressure on the corneal surface. In this regard, our results should be treated with caution. That is, postoperative topographic measurements performed after complete healing of the eyelid might have shown altered results.

Axis rotation is an important factor for the implantation of toric intraocular lens, especially bifocal or trifocal lens. Patients in their 70s are likely to undergo cataract surgery, which is consistent with the average age of the AP group in this study. Numerous patients undergoing cataract surgery have corneal astigmatism and are eligible for toric lens implantation to restore the vision quality [32,33]. When planning cataract surgery, it is ideal to use corneal topography to identify whether the eyelids affect the cornea. The astigmatic axis after blepharoptosis surgery

could interfere with the clinical improvement. Therefore, it is also important to consider the sequence of cataract surgery and blepharoptosis surgery [13].

We observed that the SIA measured using the Jaffe method was 0.78±0.70 D in the AP group and 0.82±0.88 D in the CLP group, and 25% of patients showed an SIA of >1.00 D. Several studies have reported on topographic changes after ptosis or blepharoptosis surgery. Brown et al. investigated astigmatic changes after ptosis and blepharoptosis surgery in a case series of 82 eyes and reported that 30% of patients undergoing ptosis surgery and 11% of patients undergoing blepharoptosis surgery had a transient astigmatic change of >1.00 D [16]. This result was consistent with ours. Savino et al. reported that corneal topography demonstrated a reduction in the average keratometry of 0.15±0.47˚ in 20 eyes of 17 patients with acquired ptosis using computerized topography. This study also reported that postoperative topographic maps showed a reduction of asymmetry [18]. Zinkernagel et al. reported the effects of different eyelid procedures with various degrees of dermatochalasis or ptosis on corneal topography, comparing the mean change in simulated keratometry after blepharoplasty (0.21 ± 0.20 D) and ptosis surgery (0.25 ± 0.25 D) [17]. Our results showed larger values than those previous reported, which implied that the results obtained with the use of simple subtraction is not consistent with those obtained using algebraic methods. [26] We believe that blepharoptosis SIA, much like astigmatism induced by any other surgery, should be assessed using both direction and magnitude. To the best of our knowledge, this is the first study to report on SIA after blepharoptosis surgery using the "Jaffe vector analysis," which is based on the formula for calculating the resultant cylinder when the optical cylinder is crossed.

There are some limitations to this study. We did not perform a long-term follow-up. Holck et al. reported that the increased WTR at 6 weeks decreased by 12 months [14]. Gingold et al. reported no statistically significant refractive change 6 months after surgery in patients with acquired ptosis [15]. Moreover, as mentioned above, the eyelid swelling might have been one of the factors affecting this postoperative result, and the result may change after complete healing of the eyelid. These ~~results~~ findings indicate that a long-term follow-up is required to investigate the axis change after blepharoptosis surgery.

To conclude, we observed that ATR was most common in age-related ptosis, and WTR was most common in contact lens-related ptosis. Re-positioning of the upper eyelid may affect visual functions due to astigmatic changes. Therefore, it is important to consider the eyelid position prior to cataract surgery for the best refractive outcome. Further, the cornea was dynamically affected by ptosis both pre- and postoperatively, and the ways in which early postoperative changes and long-term astigmatic changes affect the quality of vision should be taken into consideration.

## Supporting information

**S1 STROBE checklist. STROBE statement.** Checklist of items that should be included in reports of observational studies.
(DOC)

**S1 Dataset.**
(PDF)

## Author Contributions

**Conceptualization:** Kazuhiko Dannoue, Keiko Uemura, Anna Takei, Tomohiko Usui.

**Data curation:** Kazuhiko Dannoue, Seiji Takagi, Keiko Uemura, Anna Takei.

**Formal analysis:** Seiji Takagi.

**Investigation:** Seiji Takagi, Anna Takei.

**Methodology:** Kazuhiko Dannoue, Seiji Takagi, Keiko Uemura, Anna Takei.

**Project administration:** Seiji Takagi, Tomohiko Usui.

**Supervision:** Tomohiko Usui.

**Writing – original draft:** Kazuhiko Dannoue.

**Writing – review & editing:** Kazuhiko Dannoue.

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
