## [Decision Letter · Decision Letter 0]

30 Jun 2021

PONE-D-21-13875

Characteristics of astigmatism before and after blepharoptosis surgery in patients with acquired ptosis

PLOS ONE

Dear Dr. Dannnoue,

Thank you for submitting your manuscript to PLOS ONE. After careful consideration, we feel that it has merit but does not fully meet PLOS ONE’s publication criteria as it currently stands. Therefore, we invite you to submit a revised version of the manuscript that addresses the points raised during the review process.

ACADEMIC EDITOR:

The manuscript is novel and well-written. It will benefit from some meticulous grammatical revision. The figures need some improvement and you might consider adding topographic maps with differences for better illustration.

We look forward to receiving your revised manuscript.

Kind regards,

Ahmed Awadein, MD, Ph.D, FRCS

Academic Editor

PLOS ONE

Journal Requirements:

"NO The funders had no role in study design, data collection and analysis, decision to publish, or preparation of the manuscript.."

3. In your Data Availability statement, you have not specified where the minimal data set underlying the results described in your manuscript can be found. PLOS defines a study's minimal data set as the underlying data used to reach the conclusions drawn

 in the manuscript and any additional data required to replicate the reported study findings in their entirety. All PLOS journals require that the minimal data set be made fully available. For more information about our data policy, please see http://journals.plos.org/plosone/s/data-availability.

"Upon re-submitting your revised manuscript, please upload your study’s minimal underlying data set as either Supporting Information files or to a stable, public repository and include the relevant URLs, DOIs, or accession numbers within your revised

 cover letter. For a list of acceptable repositories, please see http://journals.plos.org/plosone/s/data-availability#loc-recommended-repositories. Any potentially identifying patient information must be fully anonymized.

Important: If there are ethical or legal restrictions to sharing your data publicly, please explain these restrictions in detail. Please see our guidelines for more information on what we consider unacceptable restrictions to publicly sharing data: http://journals.plos.org/plosone/s/data-availability#loc-unacceptable-data-access-restrictions.

 Note that it is not acceptable for the authors to be the sole named individuals responsible for ensuring data access.

Reviewers' comments:

Reviewer's Responses to Questions

**Comments to the Author**

1. Is the manuscript technically sound, and do the data support the conclusions?

Reviewer #1: Yes

Reviewer #2: No

Reviewer #3: Yes

2. Has the statistical analysis been performed appropriately and rigorously? 

Reviewer #1: Yes

Reviewer #2: Yes

Reviewer #3: Yes

3. Have the authors made all data underlying the findings in their manuscript fully available?

Reviewer #1: Yes

Reviewer #2: No

Reviewer #3: Yes

4. Is the manuscript presented in an intelligible fashion and written in standard English?

Reviewer #1: Yes

Reviewer #2: No

Reviewer #3: Yes

5. Review Comments to the Author

Reviewer #1: In this study, the authors investigated the change of corneal astigmatism before and after ptosis surgery in patient with acquired ptosis. The subjects were divided into two groups; age related ptosis and contact lens related ptosis. The authors compared the characteristics of astigmatism after ptosis surgery between two groups. The results were interesting and worth reporting.

The study is well-designed, and the manuscript is clear and well-written.

Reviewer #2: 1. Data are available from the Dannnoue Eye Clinic Institutional Data Access. Not clear whether the general public will be able to access the data from the institution without restrictions.

2. What is the justification for using the Crave/Jaffe method to calculate astigmatism?

3. Would be worthwile to discuss the differences noted between the Jaffe result and the Cravy results.

4. Fig 2. A and Fig 2B are hard to make sense of and don’t communicate much to the reader.

5. The authors may benefit from making more cross references to similar works. Detailed comparisons of their findings and thoughts to existing publications to either support or disregard their findings will enhance their article.

6. “This result indicates that the relationship between eyelid astigmatism and the severity of ptosis may not be linear but multifactorial. We speculate that one of the factors affecting this may be postoperative eyelid swelling. The eyelids are histologically prone to swelling and, sometimes, there can be a significant swelling after ptosis, which can suppress the corneal surface.” This suggests that the post-operative topography measurements were done before the eye had completely healed. The study method can therefore be improved by comparing post-operative SIA when the eye is completely healed to pre-operative SIA so that its really like for like. Incorrect claims may be made from not comparing like for like. As it is a retrospective study the authors may have access to data from a later review date of the patients.

7. Great point to consider lid position prior to cataract surgery for best refractive outcome.

8. Fig 3 and fig 4 have the same caption but are clearly different. Not sure they are necessary as they do not enhance the readers understanding of the results.

9. Editorial help may be needed as the case being made by the writers is hard to follow. A number of grammatical errors were noted through-out the paper. eg page 6 All examinations were performed using the upper eyelid opened by the examiner. The type of astigmatism type was divided according to the axis of astigmatism

10. Results: 57 patients (AP group: 85 eyes from 45 patients, CLP group: 23

28 eyes from 13 patients): 45+13=58 not 57 unless one patient had both AP and CLP. This will affect most results quoted

11. Page 5: Review definition of MRD as it relates to the cornea rather than the pupil as stated by authors

Reviewer #3: I read with great interest your study and I found it of high scientific value

I recommend that you could include topographic data and difference maps to show the effect of the surgery on corneal astigmatism

6. PLOS authors have the option to publish the peer review history of their article (what does this mean?). If published, this will include your full peer review and any attached files.

Reviewer #1: No

Reviewer #2: No

Reviewer #3: **Yes: **Mohamed Tarek Elnaggar

---

## [Author Response · Author response to Decision Letter 0]

4 Aug 2021

Reviewer1

> #1: In this study, the authors investigated the change of corneal astigmatism before and after ptosis surgery in patient with acquired ptosis. The subjects were divided into two groups; age related ptosis and contact lens related ptosis. The authors compared the characteristics of astigmatism after ptosis surgery between two groups. The results were interesting and worth reporting.

The study is well-designed, and the manuscript is clear and well-written.

Response: 

Thank you for your appreciative comment. We think that pre- and postoperative refractive change could be an important factor in patients’ complaints regarding visual function. We are very grateful for this opportunity to have our report published in PLOS ONE.

Reviewer #2: 

1. Data are available from the Dannnoue Eye Clinic Institutional Data Access. Not clear whether the general public will be able to access the data from the institution without restrictions.

We apologize for the missing data set in the first draft of the manuscript. We were unaware that it is always required for submission. 

The data set has been included in a supporting information file (S1) and a caption has been included at the end of the manuscript.

2. What is the justification for using the Crave/Jaffe method to calculate astigmatism?

Thank you for your question.

Several previous studies that have reported on the refractive change following blepharoptosis surgery have mentioned the use of simple subtraction, which is not consistent with results obtained using algebraic methods. 

In previous studies on surgical induced astigmatism (SIA), especially that following cataract surgery, several methods for calculating SIA have been reported by Jaffe and Clayman1, Cravy2, Holladay3, Olsen4, Naeser5, and others. 

The polar value methods are conceptually based on the surgically induced flattening and torque of the preoperative cylinder6,7, and most of these studies arrived at the same values through different approaches8.

Therefore, we chose one of the original calculation methods; i.e., the “Jaffe vector analysis”.

However, the methods developed by Cravy and Naeser must be distinguished from the others, since assessments using these calculations are associated with the axial variation rather than the cylindrical. In these methods, the change in axis is a major factor, which was the focus of our study. 

The Cravy method summarizes the magnitude and direction of SIA in a manner that is simple and easy for the readers to understand. Therefore, we chose the Cravy method to investigate whether the patients showed the WTR shift or ATR shift following blepharoptosis. 

We have included the justification for choosing the Cravy and Jaffe methods in the manuscript.

1. Jaffe NS, Clayman HM. The pathophysiology of corneal astigmatism after cataract extraction. Trans Am Acad Ophthalmol Otolaryngol 1975; 79:OP615–OP630

2. Cravy TV. Calculation of the change in corneal astigmatism following cataract extraction. Ophthalmic Surg 1979; 10:38–49

3. Holladay JT, Cravy TV, Koch DD. Calculating the surgically induced refractive change following ocular surgery. J Cataract Refract Surg 1992; 18:429–443

4. Dam-Johansen M, Olsen T, Theodorsen F. The longterm course of the surgically-induced astigmatism after a scleral tunnel incision. Eur J Implant Refract Surg 1994;

6:337–343

5. Naeser K. Conversion of keratometer readings to polar values. J Cataract Refract Surg 1990; 16:741–745

6. Naeser K Popperian falsification of methods of assessing surgically induced astigmatism Cataract Refract Surg . 2001 Jan;27(1):25-30. doi: 10.1016/s0886-3350(00)00605-2.

7. Noel A. Alpins, FRACO, FRCOphth, FACS Vector analysis of astigmatism changes by flattening, steepening, and torque J Cataract

Refract Surg 1997; 23:1503-1514

8. P J Toulemont Multivariate analysis versus vector analysis to assess surgically induced astigmatism J Cataract Refract Surg.1996 Sep;22(7):977-82. doi: 10.1016/s0886-3350(96)80203-3.

3. Would be worthwhile to discuss the differences noted between the Jaffe result and the Cravy results.

Thank you for your suggestion.

As mentioned above, the Jaffe vector analysis and the Cravy method utilize different concepts.

Jaffe and Clayman used rectangular and polar coordinates to determine, using vector analysis, the formula for calculating SIA and its axis using preoperative and postoperative corneal astigmatism.1,2

In contrast, the Cravy method utilizes a mathematical concept that is not based on optics.

We have emphasized these differences in the Discussion section. 

1. Jaffe NS, Clayman HM. The pathophysiology of corneal astigmatism after cataract extraction. Trans Am Acad Ophthalmol Otolaryngol 1975; 79:OP615–OP630

2. Alpins NA. A new method of analyzing vectors for changes in astigmatism. J Cataract Refract Surg. 1993 Jul;19(4):524-33. doi: 10.1016/s0886-3350(13)80617-7. PMID: 8355160

3. P J Toulemont Multivariate analysis versus vector analysis to assess surgically induced astigmatism J Cataract Refract Surg.1996 Sep;22(7):977-82. doi: 10.1016/s0886-3350(96)80203-3.

4. Fig 2. A and Fig 2B are hard to make sense of and communicate much to the reader.

Thank you for your comment.

In this study, we were mainly interested in the postoperative changes in the axis and we considered that the figures would allow the readers to comprehend the axis change more easily. However, what is shown in Fig. 2 is essentially the same as what is shown in Table 2, and we believe that considering surgery-induced astigmatism as a vector was not consistent with the purpose of this study.

Considering only the axis as the magnitude of astigmatism is clinically insignificant.1

We have decided to remove this figure as we believe that it may convey wrong information to the readers, since the astigmatic change was assessed only in terms of the axis change.

1. Olsen T, Dam-Johansen M. Evaluating surgically induced astigmatism. J Cataract Refract Surg. 1994 Sep;20(5):517-22. doi: 10.1016/s0886-3350(13)80231-3.

5. The authors may benefit from making more cross references to similar works. Detailed comparisons of their findings and thoughts to existing publications to either support or disregard their findings will enhance their article.

Thank you for your comments. 

We have referred to previous papers, mainly on the refractive change after ptosis surgery; however, we found few citations for original studies that have assessed induced astigmatism following cataract surgery and have added these references (listed in response to question No. 2) to enrich the manuscript.

 6. This result indicates that the relationship between eyelid astigmatism and the severity of ptosis may not be linear but multifactorial. We speculate that one of the factors affecting this may be postoperative eyelid swelling. The eyelids are histologically prone to swelling and, sometimes, there can be a significant swelling after ptosis, which can suppress the corneal surface. This suggests that the post-operative topography measurements were done before the eye had completely healed. The study method can therefore be improved by comparing post-operative SIA when the eye is completely healed to pre-operative SIA so that its really like for like. Incorrect claims may be made from not comparing like for like. As it is a retrospective study the authors may have access to data from a later review date of the patients.

You have raised a very valid point, which is a major limitation of this study, as stated in the manuscript.

Eyelids are inherently prone to swelling, and this might have been a major cause for corneal deformity at 1 month postoperatively. It would be more scientific to compare the results using data that was obtained long after the surgery, i.e., once the effects of eyelid swelling had worn out.

To the best of our best knowledge, the ways in which corneal astigmatism changes over time following ptosis surgery remains inconclusive.

Holck et al. reported that the WTR increased at 6 weeks and decreased at 12 months. In contrast, Ingold et al. reported that there was no statistically significant change in the refractive error at 6 months postoperatively in patients with acquired ptosis. These have been included in the manuscript (Line Page)

Since the results of this study were limited to short-term changes observed after 1 month, the title of this paper has been revised to "Characteristics of astigmatism before and 1 month after blepharoptosis surgery in patients with acquired ptosis” to avoid misleading the reader that these results are permanent.

Further, we have acknowledged the possibility that the results might have changed after complete healing of the eyelid. 

During the early postoperative period, patients are very anxious about changes in their vision, and we believe that our results associated with the changes observed 1 postoperative month are still scientifically relevant.

Unfortunately, several patients who participated in the study dropped out before the postoperative follow-up, and we were unable to obtain sufficient data that was statistically significant; therefore, we limited our findings to changes observed at 1 month postoperatively.

We strongly believe that a long-term follow-up study is necessary, and we would be interested in observing the longitudinal changes in SIA following cataract surgery. We are currently following up new cases with prospective 1-year data.

7. Great point to consider lid position prior to cataract surgery for best refractive outcome.

Thank you for your appreciative comments. We also believe that consideration of the eyelid position prior to cataract surgery is significantly important to achieve the best refractive outcome following the surgery, which was one of the main focus areas of this study.

We have emphasized this point in the conclusion of the manuscript.

8. Fig 3 and fig 4 have the same caption but are clearly different. Not sure they are necessary as they do not enhance the readers understanding of the results.

Thank you for pointing this out. We have made the correction in Fig. 3.

 9. Editorial help may be needed as the case being made by the writers is hard to follow. A number of grammatical errors were noted through-out the paper. eg page 6 All examinations were performed using the upper eyelid opened by the examiner. The type of astigmatism type was divided according to the axis of astigmatism

Thank you for your comment.

We have sought help from a professional English editing service and re-checked the entire text.

All examinations were performed using the upper eyelid opened by the examiner.

→To avoid any pressure from the eyelids on the corneal surface, the examination was performed with the patients’ eyes open.

The type of astigmatism type was divided according to the axis of astigmatism

→We have deleted this sentence.

10. Results: 57 patients (AP group: 85 eyes from 45 patients, CLP group: 23

> 28 eyes from 13 patients): 45+13=58 not 57 unless one patient had both AP and CLP. This will affect most results quoted

Thank you for pointing this out.

This was a simple calculation error; the number of patients in the AP group was 58 and not 57.

I have included the data set in a supporting file.

11. Page 5: 

Review definition of MRD as it relates to the cornea rather than the pupil as stated by authors

Thank you for pointing this out.

We have reviewed the definition of MRD as stated by other authors and have revised it as the measurement of the distance from the middle upper eyelid to the corneal light reflex.

Reviewer #3: I read with great interest your study and I found it of high scientific value I recommend that you could include topographic data and difference maps to show the effect of the surgery on corneal astigmatism

Thank you for your appreciative comments. We have added a topographic map of the two representative cases in Fig. 4. 

Case 1: Postoperative topography showing change in axis with improvement in focal steepening in the superior corneal hemisphere; Case 2: Postoperative topography showing increased magnitude of astigmatism and decreased uncorrected visual acuity 1 month postoperatively.

---

## [Editor Report · Decision Letter 1]

4 Oct 2021

Characteristics of astigmatism before and 1 month after blepharoptosis surgery in patients with acquired ptosis

PONE-D-21-13875R1

Dear Dr. Dannnoue,

We’re pleased to inform you that your manuscript has been judged scientifically suitable for publication and will be formally accepted for publication once it meets all outstanding technical requirements.

Kind regards,

Ahmed Awadein, MD, Ph.D, FRCS

Academic Editor

PLOS ONE
---

## [Editor Report · Acceptance letter]

6 Oct 2021

PONE-D-21-13875R1 

Characteristics of astigmatism before and 1 month after blepharoptosis surgery in patients with acquired ptosis 

Dear Dr. Dannoue:

I'm pleased to inform you that your manuscript has been deemed suitable for publication in PLOS ONE. Congratulations! Your manuscript is now with our production department. 

Kind regards, 

on behalf of

Dr. Ahmed Awadein 

Academic Editor

PLOS ONE